# Why Don’t We Tube Feed Hip Fracture Patients? Findings from the Implementation of an Enteral Tube Feeding Decision Support Tool

**DOI:** 10.3390/geriatrics6010012

**Published:** 2021-02-02

**Authors:** Sally Barrimore, Madeleine Davey, Ranjeev Chrysanth Pulle, Alisa Crouch, Jack J. Bell

**Affiliations:** 1The Prince Charles Hospital, Chermside, QLD 4032, Australia; Sally.Barrimore@health.qld.gov.au (S.B.); Chrysranjeev.pulle@health.qld.gov.au (R.C.P.); Alisa.Crouch@health.qld.gov.au (A.C.); 2School of Exercise and Nutrition Sciences, Queensland University of Technology, Kelvin Grove, QLD 4059, Australia; M.Davey@qut.edu.au; 3School of Human Movement and Nutrition Sciences, The University of Queensland, St. Lucia, QLD 4067, Australia

**Keywords:** clinical decision-making, enteral nutrition, geriatrics, hip fracture, malnutrition

## Abstract

Background: This study aimed to report (i) the prevalence of enteral tube feeding (ETF), (ii) investigate whether implementing a decision support tool influenced ETF rates, and (iii) understand reasons influencing decisions to offer ETF. Methods: A pre/post evaluation included consecutive patients admitted to a hip fracture unit. Following baseline data collection, a published ETF Decision Support Tool was implemented by the multidisciplinary team to determine the necessity and influencing reasons for offering ETF. Results: Pre-post groups (*n* = 90,86) were well matched for age (83 vs. 84.5 years; *p* = 0.304) and gender (females 57 vs. 57; *p* = 0.683). ETF rates remained low across groups (pre/post *n* = 4,2; *p* = 0.683) despite high malnutrition prevalence (41.6% vs. 50.6%; *p* = 0.238). Diverse and conflicting reasons were identified regarding decisions to offer ETF. Conclusion: A complex interplay of factors influences the team decision-making process to offer ETF to nutritionally vulnerable patients. These demands are individualised, rather than algorithmic, involving shared decision-making and informed consent processes.

## 1. Introduction

Acute hip fracture patients are an especially vulnerable population. Malnutrition is recognised as an independent predictor of 12-month mortality, delayed recovery and contributes to increased health care costs [1,2,3]. More than half of hip fracture patients are malnourished, with a substantial proportion of patient’s nutritional status deteriorating during admission due to poor protein and energy intake, despite post-operative multidisciplinary nutrition care [4].

Evidence-based guidelines for the management of malnutrition in hospital inpatients recommend that if a patient cannot maintain oral intake of more than 50%, enteral tube feeding (ETF) should be considered [5,6]. Despite a high reported prevalence of malnutrition and inadequate oral intakes in hip fracture populations, the efficacy of ETF in this patient population remains unclear. Australian and international guidelines for hip fracture care acknowledge the importance of appropriate nutrition but do not provide any clear direction regarding tube feeding [7]. The 2016 Cochrane review calls for further research regarding the use of enteral tube feeding in very malnourished hip fracture patients [8]. 

In the absence of high-level evidence or recommendations regarding if and when to commence ETF in hip fracture patients, the decision to use enteral nutrition options should be made between the patient (and their carers) and their treating team in a shared decision process. A recent study highlighted the complex interplay of factors relating to patient acceptance of ETF related to knowledge and understanding, patient refusal and necessity of the tube as key patient and carer barriers to implementing enteral tube feeding in hip fracture patients [9]. In the absence of a supporting evidence base, determining what care should be provided may be considered using a two-step approach. Firstly, clinical judgement with consideration by the treating team regarding whether tube feeding in any individual may be of benefit, futile, or even harmful; and then consequently applying shared decision making and informed consent processes to ensure care is provided in line with the patient’s treatment goals and preferences. Mon et al., (2017) developed the Enteral Tube Feeding Decision Support tool (ETFDST) via a modified Delphi approach across Australian and New Zealand hip fracture sites to support clinicians and patients as the first step of this decision-making process [10]. This tool supports the multidisciplinary team to make informed decision-making processes through considering key clinical factors that influence whether feeding may or may not be indicated. A non-algorithmic, clinical judgement decision is then able to be made regarding whether or not tube feeding should be offered to an individual patient or carer as an appropriate treatment option for consideration [10,11].

This quality assurance activity consequently aimed to ascertain whether the implementation of the ETFDST influenced rates of ETF, and to highlight the factors influencing clinician recommendations for tube feeding.

## 2. Materials and Methods

The study is a pre/post pragmatic action cycle evaluation of changes to clinical practice performed at a tertiary hospital’s orthogeriatric ward [12]. An ethics exemption was approved by our institution’s Human Research Ethical Committee and was approved on 30 January 2019.

### 2.1. Study Participants and Protocol 

All patients with a confirmed neck or femur fracture on imaging admitted to the orthogeriatric ward were eligible for participation in the study. Patients were excluded if they were for conservative management; goals of care identified as palliative perioperatively; aged <18 years of age; died before a nutrition assessment could be carried out or refused any dietetic input. Residential aged care facility residents are often not considered for ETF in the unit due to a shorter length of stay. Consequently, whilst community-dwelling and residential aged care home residents were recruited as participants, the a priori power calculation was undertaken with a confidence level of 95% and a power level of 0.8 to detect a clinically significant change in the number of community-dwelling patients being offered an informed consent decision regarding ETF. This assumed an estimated baseline of 10% with a predicted increase of tube feeding rates to 30%. The suggested sample size was consequently 59 community-dwelling patients (in both the pre- and post-implementation groups).

### 2.2. The Enteral Tube Feeding Decision Support Tool

The ETFDST was developed by a modified Delphi process to facilitate a timely, informed, team decision-making process to determine if ETF is likely to be in the individual patient’s best interest. The ETFDST poses a series of nine questions (outlined in Table 1) to be completed by the multidisciplinary team using a subjective rating scale (highly likely to highly unlikely). This is a subjective tool; not all elements need to be completed for each patient to inform the treating teams’ decision-making. The tool is consequently reliant on clinical judgement and there are no ranking, quantitative evaluation, or algorithmic processes embedded within the tool.

### 2.3. Implementation

The ETFDST was implemented in April 2019. The launch utilised key stakeholders and opinion leaders involved in the tool development with marketing material and education through team meetings, case conferences, and local clinical governance meetings. Process measures targeted periodic review of implementation to identify the feasibility and fidelity of tool application, and any iterative changes to the process required [13]. Completion was undertaken by the treating dietitian and/or geriatrician during twice-weekly clinical case conferences and considered input from routinely attending multidisciplinary healthcare workers including geriatric and medical specialists and trainees, nursing staff, dietitians, occupational therapists, physiotherapists, speech pathologists, social workers, and students. The final decision regarding whether to offer enteral tube feeding was made by the treating geriatrician. In line with the routine practice for this unit, hip fracture patients received a routine nutritional assessment and intervention by a dietitian and treating team. This included the routine provision of; high protein/energy diets, prescribed medical nutrition supplements, interdisciplinary education regarding the role of nutrition following hip fracture surgery, and discharge planning considering nutrition care [14].

### 2.4. Data Collection

Predicted inadequate protein/energy intake was defined as an expected intake of less than 50% of energy or protein requirements in line with local and international treatment guidelines [5,6]. Requirements were estimated by the ward dietitian applying the ratio method for hip fracture patients in the post-operative period of 125 kJ/kg and 1.2 g/kg protein; assessment for a diagnosis of predicted inadequate intake was undertaken and documented in the early post-operative period by the treating dietitian as per the routine clinical nutrition model of care [5,6,14]. Malnutrition was routinely assessed by treating dietitians using the Subjective Global Assessment and/or ICD10-AM coding criteria as per routine clinical practice [14]. These data and other metrics were collected from routine multidisciplinary team documentation in a patient’s medical records. Data were collected by two independent reviewers. A sample of 10 charts was audited by both reviewers to ensure interrelated reliability. Responses to each question on the ETFDST were classified according to the criteria on the tool as highly likely, neutral, highly unlikely and not completed.

### 2.5. Statistical Analysis

The analysis was performed using the Statistical Package for Social Sciences (SPSS Version 25). Between-group demographic comparisons were made for age using the Mann-Whitney U-test as data was not normally distributed. Associations and comparisons were made between categorical variables of interest (gender, from a private residence, malnutrition) using a Chi-squared or Fisher’s exact test where appropriate. The criterion for statistical significance was set at the conventional level of *p* < 0.05 (two-tailed) for all analyses.

## 3. Results

Data were available for 90 patients pre-implementation (July–November 2018) and 86 patients post-implementation (April–July 2019). One hundred and seventy-six patients were included in the analysis. Groups were comparable demonstrating an older, female-biased sample that tended to be admitted from private residence (Table 2). Malnutrition rates were high in both samples, and although not statistically different, there was a higher prevalence observed in the post-implementation group.

The ETFDST was implemented into routine clinical practice by the treating team. Initially, the goal was to complete the tool on all admitted patients, however, the demand for the completion of the tool for patients who were not at nutrition risk was considered to be outweighed by the acceptability and practicality of completing the tool for nutritionally vulnerable patients. Therefore, the treating team adapted the fidelity criteria to complete ETFDST on patients who were predicted to meet <50% of their estimated energy and protein requirements in the following 5 days rather than on all patients. Consequently, the ETFDST was completed as part of a multidisciplinary case conference on 20 patients that were predicted to not meet protein and energy requirements within 5 days. These patients were more likely to also have a diagnosis of malnutrition (*n* = 15/20) than the overall cohort. 

There were no significant or clinically relevant differences in the number of patients receiving ETF in the pre and post-implementation groups with 4 patients and 2 patients respectively, receiving ETF (*p* = 0.683). Following ETFDST completion, only 3 of 20 patients (15%) were deemed appropriate to undertake an informed consent discussion between the treating team and patient and carers to consider ETF as a treatment option. Of these patients, two received enteral feeding and one declined ETF intervention. This left 17/20 (85%) patients with predicted inadequate intake who were not deemed appropriate to offer an enteral tube. Documented ETFDST categories for not offering an ETF in patients diagnosed with predicted inadequate protein and energy intake are highlighted in Table 3.

The 3 patients deemed appropriate to undertake an informed consent discussion following ETFDST completion were predicted to meet < 50% of their estimated energy and protein requirements in the next 5 days with other reasons listed in Table 4. The average age of these patients was 83 years.

## 4. Discussion

To the authors’ knowledge, this study is the first to implement and evaluate a decision support tool to facilitate clinical decision-making regarding ETF recommendations in hip fracture. Findings also highlight that ETF is rarely offered to hip fracture patients (or their carers) with predicted inadequate protein and energy intake, and in most cases, concurrent malnutrition. Implementing a clinician-led ETFDST to support ETF use did not influence ETF rates; these remained low pre and post-implementation. Our findings are also the first to articulate the reasons from a treating team perspective for why ETF may not be offered to nutritionally vulnerable hip fracture patients.

A Cochrane review suggests that in the absence of supporting literature, trials of ETF in hip fracture patients should be limited to those with severe malnourishment [8]. However, prior studies have demonstrated ongoing poor ongoing oral intake in patients admitted to hospital with hip fractures leading to further nutritional deterioration during admission [4]. Hospital-acquired malnutrition delays discharge, independently increases morbidity and mortality and incurs substantial treatment costs and penalties in the Australian setting [1,2,3]. Inducing likely irreversible sarcopaenia in multimorbid, older, nutritionally vulnerable patients through delaying nutritional intervention until severe malnutrition develops is consequently unlikely to represent the best interests of patients, nor their treating teams [2]. Whilst a prior study targeting multidisciplinary, multimodal nutrition care processes has demonstrated increase intake and consequent reduced inpatient nutritional decline post-implementation, there remained a considerable proportion of patients with inadequate intake and nutritional deterioration in hospital in the absence of increased enteral tube feeding usage [14].

In this study, identification and documentation of inadequate intake and malnutrition risk were undertaken for the entire cohort. However, findings demonstrate these are not by themselves predicting factors for recommending tube feeding in this patient cohort. For three in four patients who were both malnourished with predicted inadequate protein and energy intake, there were other clinical elements considered that precluded discussing ETF with the patient as a viable treatment option. This highlights a complex interplay of factors affecting clinical judgement to offer and recommend ETF, particularly in the absence of supportive efficacy data for tube feeding in this population [8,15,16]. The patient’s ability to take medications orally (*n* = 14) was considered in many cases as a limiting factor for ETF use. Previous studies have highlighted that malnutrition needs to be treated as a disease not without consequence, in which food needs to be considered as medicine after acute hip fracture [4]. This raises an interesting point related to the relative importance placed on nutrition support as a medication or as a food.

Risk of repeated dislodgement of the nasogastric tube (NGT) (*n* = 11), NGT-related contraindications or complications (*n* = 8) and NGT aspiration risk (*n* = 6) were sighted as a barrier to offering ETF. Poor ETF tolerance has been widely reported in the literature (ie: tube intolerance, diarrhoea) but no adverse outcomes (i.e.: death, complications) have been clearly articulated [8,17,18]. Medical contraindications such as severe dementia or palliation (*n* = 11) were present as barriers for several patients. Evidence for the use of ETF in the cognitively impaired remains scarce. The American Geriatrics Society’s position statement insists that careful hand feeding is almost as good as tube feeding for outcomes of comfort, aspiration pneumonia, function status and death while at the same time avoiding the burden and complications associated with tube feeding in those with cognitive impairment [19,20]. Results highlight the importance of considering patients’ expressed or documented wishes related to tube feeding (*n* = 4). The hip fracture population tends to be older and a multimorbid patient group requiring consideration of multiple factors to ensure the intervention is in the patients’ best interests [15,16,21]. Five patients with predicted inadequate intake were well-nourished; it is prudent to consider that prolonged inadequate nutrition to well-nourished patients in the post-operative period is likely to result in iatrogenic malnutrition [14]. The authors would suggest a reconsideration of calls to reserve enteral tube feeding for trials in those who are ‘very malnourished’ [8].

This study has highlighted a broader question related to what to do when the evidence base is unclear and who needs to be involved in this decision-making process. Bringing the discussion back to the question—is ETF largely inappropriate in the hip fracture population and does the risk out way the benefit?

The answer to this question is complex and is unable to be addressed by applying algorithmic processes or population-based recommendations for multimorbid, older patient populations [9,10]. This is particularly relevant in hip fracture, where evidence for overall benefits versus the risk for multiple and sometimes conflicting interventions remains unclear [8,18]. As a case example, the degree of risk associated with the insertion of a temporary access enteral tube may be clearly increased in an individual with known oesophageal varices. However, if this patient has profound malnutrition, severe dysphagia and drowsiness and a need to take oral medications for the management of likely encephalopathy, this decision becomes increasingly difficult. The balance of risk in this situation can be difficult to quantify and may need to be considered along with the patient’s nutrition status, level of swallowing dysfunction and the need to take oral medications. These reasons alone must be viewed through the lens of patient treatment goals, perceived quality of life, and what is in the individual patient’s best interest [9,16,22]. For this reason, we would advocate that whilst an algorithm may be entirely inappropriate, a subjective decision support tool may facilitate clinical decision making.

With only three patients offered ETF in this study, our findings are inadequate to draw meaningful conclusions regarding reasons considered to support offering ETF (Table 4). Our results do, however, demonstrate that patients routinely did not have a strong, singular risk to preclude offering ETF (Table 3 and Table 4). In all cases, several indicators of potential adverse outcomes were present. In some cases, one strong indicator may have highly impacted the team discussions regarding whether to offer ETF or not. In other individuals, multiple lower ‘impact’ indicators may have been present. Consequently, a simple ‘count’ of support tool domains is not appropriate [10]. In recognition of this complexity, the authors do not support suggestions that tube feeding in hip fracture patients should be reserved for those most malnourished, and would go so far as to suggest these are less likely to benefit from ETF than those with, or at risk of malnutrition who have active rehabilitation intent. Support for maximising nutritional intake should continue in such cases where dedicated attempts by the interdisciplinary team to maximise oral intake remain inadequate. This tool can be used as a prompt for discussion within the team and offers a mechanism to clarify concerns prior to discussions with patients and families.

The complex and invasive nature of ETF and the lack of supporting efficacy evidence makes this a complex decision-making process. Adding to this complexity, when the treating team does recommend ETF as a potential treatment option, are a series of factors that influence a patient’s decision to accept ETF or not [9,22]. King et al., (2017) explored ETF preferences with patients and carers, finding that most patients do not perceive EFT to be a treatment option they would likely accept. A number of complex interrelationships of factors impacted patients’ and caregivers’ ETF acceptance and refusal of ETF including knowledge and understanding, perceived consequences and the necessity of the tube [9]. This tool can be used to support treating teams in considering the complex nature of tube feeding to ensure patients receive clear information about their treatment options to undertake an informed decision-making process.

The ETFDST was completed on a small number of hip fracture patients with predicted inadequate oral intake and is a limitation of this study, noting though that this study was conducted over a 2-year timeframe. This study presents a single site perspective although the implementation ward has a demonstrated strong MDT nutrition-focused model of care 14. A further limitation was the overestimate of baseline tube feeding rates; ETF rates for the total sample size (2%) remained similar to that previously reported on the orthogeriatric ward during a 2012 study [2]. Whether the sample size estimated using a priori power calculation was adequate is a consequent additional limitation. Implementation groups matched and were similar to previous studies in this population [14] The use of the ETFDST, previously validated through a modified Dephi process is a strength of this study, the subjective nature of the decision support tool allows for the treating team to consider each factor prior to making a global clinical judgement decision regarding whether enteral tube feeding should be considered as meritorious or futile [10].

## 5. Conclusions

Our findings clearly articulate multiple reasons why nutritionally vulnerable hip fracture patients may not be offered ETF. Applying a clinical decision support tool to identify these did not appear to influence tube feeding rates. Why patients do not receive tube feeding does not appear directly attributable to individual barriers or enablers to tube feeding; instead, a complex interplay of factors is involved. We, therefore, conclude that until there is clear evidence to support the efficacy of tube feeding in hip fracture patients, tube feeding rates are likely to remain low. Whether or not ETF is considered in the patients’ best interests or futile should not be predicated on single, quantifiable or algorithmic responses. However, any process that supports treating teams to carefully consider the key barriers and enablers to tube feeding in hip fracture should be considered meritorious. This will ensure a balanced risk approach between optimising informed consent and shared decision making, food as a medicine versus food for comfort, and avoiding the unnecessary burden of offering futile or harmful care.

## Figures and Tables

**Table 1 geriatrics-06-00012-t001:** Enteral Tube Feeding Decision Support Tool, questions [10].

Well-nourished Able to meet estimated protein/energy requirements within 5 days NGT-related contraindication or complications Risk of repeated dislodgement of NG tube NGT-related aspiration risk Able to take required medications per oral Against patients expressed or documented wishes Medically contraindicated (eg: Advanced dementia, palliative) Other

NGT: Nasogastric tube.

**Table 2 geriatrics-06-00012-t002:** Demographic Details of study patients.

Demographic Characteristics	Pre-Implementation	Post-Implementation	Statistic
Age (years) median	83.0	84.5	*p* = 0.304 ^†^
Gender (female) n (%)	57 (63%)	57 (66%)	χ^2^ (1) 0.167; *p* = 0.683
From private residence, n (%)	63 (70%)	57 (66%)	χ^2^ (1) 0.281; *p* = 0.596
Malnutrition, n (%)	37 (41.6%)	43 (50.6%)	χ^2^ (1) 1.423 *p* = 0.233

Malnutrition was diagnosed using the Subjective Global Assessment (SGA B or SGA C). Malnutrition diagnosis assessment data was missing for one patient in the pre cohort and one patient in the post cohort. ^†^ Mann-Whitney U test

**Table 3 geriatrics-06-00012-t003:** Enteral Tube Feeding Decision Support tool indicated barriers to ETF.

	*n*
Able to take medications per oral	14
Risk of repeated dislodgement of NGT	11
Medically contraindicated (e.g., Advanced dementia, palliative)	11
NGT-related contraindications or complications	8
NGT-related aspiration risk	6
Well-nourished	5
Against patient’s expressed or documented wishes	4
Other	0

NGT: Nasogastric tube.

**Table 4 geriatrics-06-00012-t004:** Profile of patients offered ETF.

	Patient 1 83 Years; F	Patient 2 90 Years; M	Patient 3 77 Years; M
Able to take medications per oral		Unlikely	
Medically contraindicated (e.g., advanced dementia, palliative)		Unlikely	Likely
NGT-related aspiration risk			Likely
Well-nourished	Unlikely	Likely	Unlikely

## Data Availability

The data are not publicly available due to ethical and privacy requirements; additional data may be available on request from the corresponding author.

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
