# Peer review of "Why Don’t We Tube Feed Hip Fracture Patients? Findings from the Implementation of an Enteral Tube Feeding Decision Support Tool"

_geriatrics, 2021, doi:10.3390/geriatrics6010012_

Round 1
Reviewer 1 Report
This quality management report summarizes practical experience along with the pre-post implementation of an already accepted algorithmic decision support tool in a geriatric ward to come to a decision whether tube feeding should be part of hipp fracture patients‘ therapy. The manuscript is well written and results are adequately presented. Only minor comments:
- It is reported that „the prediction to meet less than 50% of energy and protein“ is the major criteria to use ETF. How this can be „clinically judged“? Are there any „objective“ criteria available? It might also interested for the reader what 50% means in absolute terms – based on reference values? Which ones?
- Even a less intake of nutrition is predicted, several individual reasons hinder the use of ETF. What happens then? Have other measures been taken to support/increase food intake or at least to long-term follow nutrition status? Or did the support team simply accept this situation?
Author Response
Reviewer 1:
This quality management report summarizes practical experience along with the pre-post implementation of an already accepted algorithmic decision support tool in a geriatric ward to come to a decision whether tube feeding should be part of hipp fracture patients‘ therapy. The manuscript is well written and results are adequately presented.
Thank you for taking the time to review and providing opportunity to further improve this manuscript. We have highlighted responses below in italics.
Only minor comments:
- It is reported that „the prediction to meet less than 50% of energy and protein“ is the major criteria to use ETF. How this can be „clinically judged“? Are there any „objective“ criteria available? It might also interested for the reader what 50% means in absolute terms – based on reference values? Which ones?
PAGE 3 line 38: Please refer tracked changes:
Predicted inadequate protein/energy intake was defined as an expected intake of less than 50% of energy or protein requirements in line with local and international treatment guidelines 5,6. Requirements were estimated by the ward dietitian applying the ratio method for hip fracture patients in the post-operative period of 125kJ/kg and 1.2g/kg protein; assessment for a diagnosis of predicted inadequate intake was undertaken and documented in the early post operative period by the treating dietitian as per the routine clinical nutrition model of care 5,6, 14.
- Even a less intake of nutrition is predicted, several individual reasons hinder the use of ETF. What happens then? Have other measures been taken to support/increase food intake or at least to long-term follow nutrition status? Or did the support team simply accept this situation?
Please refer new text:
In line with routine practice for this unit, hip fracture patients received a routine nutritional assessment and intervention by a dietitian and treating team. This included routine provision of; high protein/energy diets, prescribed medical nutrition supplements, interdisciplinary education regarding the role of nutrition following hip fracture surgery, and discharge planning considerate of nutrition care [14].

Reviewer 2 Report
This is a single center prospective evaluation of implementing a validated assessment tool, the Enteral Tube Feeding Decision Support Tool (ETFDST), in a multidisciplinary team caring for older adults admitted to the hospital for hip fracture. The study provides insight to the barriers and challenges of tube feeding to address malnutrition, a common problem in this patient population. As the authors highlight, the lack of evidence to support the benefit of tube feeding is likely the biggest barrier to uptake. The data presented in this study may be used to inform future efficacy studies.
The paper could be strengthened by addressing the following items:
- Lines 64-66: the authors use the words “influence rates of appropriate ETF”, but is there a gold standard for when ETF should be used? It suggests that there are standard criteria for ETF but based on the literature review presented in the introduction that does not seem to be the case.
- Method section 2.1- it would be helpful to include a flow diagram presenting the number of patients meeting inclusion/exclusion criteria during the study period. Some of this detail is described in results- consider laying out participant selection in more detail in the methods.
- Please clarify whether the primary outcome is the number of patients offered ETF (and not the number of patients getting ETF). The sample size is based on a 20% increase in the number of patients being offered ETF. What is the rationale for the 20% difference? Was the 10% baseline rate based on hospital record data? The rate in the “pre intervention group” was 4%....Why do you think there will be a significant increase in offering ETF with the implementation of the assessment tool?
- Section 2.3- provide more detail about the implementation. Who is on the multidisciplinary team? Who is responsible for filling out the decision support tool and making the final decision about offering EFT? When is the tool completed? How is malnutrition assessed?
- Statistical analysis – either here or in the description of participants I would provide more detail about matching and participant selection.
- Table 2- approximately 30% of patients were not from a private residence, though I thought this was an exclusion criteria. Also- define SGA, B or C in a footnote and include sample “n” in each column for pre implementation and post implementation cohort.
- The team made the decision to only implement the tool for patients that were predicted to meet <50% of estimated energy and protein requirements. How was this determined? It seems prudent to compare to similar patients in the pre implementation cohort. How many of the 4 patients offered ETF in the pre implementation cohort got a feeding tube? If none, is there any documentation in the records explaining why?
- Table 3 is missing the “n” in the column on the right. I’m curious about the 5 “well nourished” patients. These patients were predicted to meet <50% of estimated energy and protein requirements. What explains the discrepancy?
- Were patients offered or put on other nutritional supplements to address muscle wasting and malnourishment?
Author Response
Reviewer 2:
Thanks for your detailed constructive feedback which has enabled us to further improve the manuscript. Please responses below in italics.
- Lines 64-66: the authors use the words “influence rates of appropriate ETF”, but is there a gold standard for when ETF should be used? It suggests that there are standard criteria for ETF but based on the literature review presented in the introduction that does not seem to be the case.
PAGE 2 line 26: Deleted ‘appropriate’
- Method section 2.1- it would be helpful to include a flow diagram presenting the number of patients meeting inclusion/exclusion criteria during the study period. Some of this detail is described in results- consider laying out participant selection in more detail in the methods.
COMMENT: In line with relevant study design checklists, the authors are satisfied that the methods are not the appropriate section to detail the number of patients recruited as provided in the results.
- Please clarify whether the primary outcome is the number of patients offered ETF (and not the number of patients getting ETF). The sample size is based on a 20% increase in the number of patients being offered ETF. What is the rationale for the 20% difference? Was the 10% baseline rate based on hospital record data? The rate in the “pre intervention group” was 4%....Why do you think there will be a significant increase in offering ETF with the implementation of the assessment tool?
Comment: We are unable to provide a methodologically robust comparison of the number of patients ‘offered’ enteral tube feeding as this data was not adequately documented in the medical record prior to ETFDST implementation. For example, the treating doctor may have had a brief discussion with the patient or carer at the bedside about tube feeding, however this was not documented.
PAGE 7 LINE 38: New text:
A further limitation was the overestimate of baseline tube feeding rates; ETF rates for the total sample size (2%) remained similar to that previously reported on the orthogeriatric ward during a 2012 study [2]. Whether the sample size estimated using a priori power calculation was adequate is a consequent additional limitation.
Refer existing text
PAGE 6 Line 51: With only three patients offered ETF in this study, our findings are inadequate to draw meaningful conclusions regarding reasons considered to support offering ETF
PAGE 7 line 22: paragraph beginning with ‘The complex and invasive nature of ETF’
- Section 2.3- provide more detail about the implementation. Who is on the multidisciplinary team? Who is responsible for filling out the decision support tool and making the final decision about offering EFT? When is the tool completed? How is malnutrition assessed?
PAGE 3 Line 24:
Completion was completed by the treating dietitian and/or geriatrician during twice weekly clinical case conferences and considered input from routinely attending multi-disciplinary healthcare workers routinely including geriatric and medical specialists and trainees, nursing, dietitian, occupational therapy, physiotherapy, speech pathology, so-cial work, and students. The final decision regarding whether to offer enteral tube feeding was made by the treating geriatrician.
PAGE 4 Line 3:
Malnutrition was routinely assessed by treating dietitians using the Subjective Global Assessment and/or ICD10-AM coding criteria as per routine clinical practice[14].
- Statistical analysis – either here or in the description of participants I would provide more detail about matching and participant selection.
Comment: Please note this was not matched samples but a comparison of pre and post independent samples.
PAGE 4 Line 13: Amended text:
Analysis was performed using the Statistical Package for Social Sciences (SPSS Version 25). Between group demographic comparisons were made for age using the Mann-Whitney U-test as data was not normally distributed. Associations and comparisons were made between categorical variables of interest (gender, from private residence, malnutrition) using a Chi-squared or Fisher’s exact test where appropriate. The criterion for statistical significance was set at the conventional level of P < 0.05 (two-tailed) for all analysis.
PAGE 4 LINE 23:
replaced ‘well matched’ with comparable
- Table 2- approximately 30% of patients were not from a private residence, though I thought this was an exclusion criteria. Also- define SGA, B or C in a footnote and include sample “n” in each column for pre implementation and post implementation cohort.
PAGE 2 Line 39: Amended text:
Residential aged care facility residents are often not considered for ETF in the unit due to a shorter length of stay. Consequently, whilst community dwelling and residential aged care home residents were recruited as participants, the a priori power calculation was undertaken with a confidence level of 95% and a power level of 0.8 to detect a clinically significant change in the number of community dwelling patients being offered an in-formed consent decision regarding ETF. This assumed an estimated baseline of 10% with a predicted increase of tube feeding rates to 30%. The suggested sample size was con-sequently community dwelling 59 patients (in both the pre- and post-implementation groups.
Refer changes made to table 2
n included
Footnote included:
Malnutrition diagnosed using the Subjective Global Assessment (SGA B or SGA C). Malnutrition diagnosis assessment data was missing for one patient in the pre cohort and one patient in the post cohort.
- The team made the decision to only implement the tool for patients that were predicted to meet <50% of estimated energy and protein requirements. How was this determined?
PAGE 3 line 38: Please refer tracked changes:
Predicted inadequate protein/energy intake was defined as an expected intake of less than 50% of energy or protein requirements in line with local and international treatment guidelines 5,6. Requirements were estimated by the ward dietitian applying the ratio method for hip fracture patients in the post-operative period of 125kJ/kg and 1.2g/kg protein; assessment for a diagnosis of predicted inadequate intake was undertaken and documented in the early post operative period by the treating dietitian as per the routine clinical nutrition model of care 5,6, 14.
It seems prudent to compare to similar patients in the pre implementation cohort. How many of the 4 patients offered ETF in the pre implementation cohort got a feeding tube? If none, is there any documentation in the records explaining why?
Comment: the 4 patients in the pre implementation cohort received ETF ie. had an NGT placed (vs 2 post implementation). The cohorts compared were similar as per demographics and the post implementation group was not limited to those who had the ETFDST completed as per paragraphs 2 and 3 in the results section.
- Table 3 is missing the “n” in the column on the right. I’m curious about the 5 “well nourished” patients. These patients were predicted to meet <50% of estimated energy and protein requirements. What explains the discrepancy?
COMMENT: These patients were assessed as currently well nourished ie. SGA-A, and 2. Predicted inadequate intake. This group is the group highly likely to develop iatrogenic malnutrition.
PAGE 6 Line 36 New text:
Five patients with predicted inadequate intake were well-nourished; it is prudent to consider that prolonged inadequate nutrition to well-nourished patients in the post-operative period is likely to result in iatrogenic malnutrition [14]. The authors would suggest reconsideration of calls to reserve enteral tube feeding for trials in those who are ‘very malnourished’ [8].
Were patients offered or put on other nutritional supplements to address muscle wasting and malnourishment?
PAGE 3 Line 30:
In line with routine practice for this unit, hip fracture patients received a routine nutritional assessment and intervention by a dietitian and treating team. This included routine provision of; high protein/energy diets, prescribed medical nutrition supplements, interdisciplinary education regarding the role of nutrition following hip fracture surgery, and discharge planning considerate of nutrition care [14].
